# Smartphone Distraction: Italian Validation of the Smartphone Distraction Scale (SDS)

**DOI:** 10.3390/ijerph20156509

**Published:** 2023-08-02

**Authors:** Maria Lidia Mascia, Mirian Agus, Łukasz Tomczyk, Natale Salvatore Bonfiglio, Diego Bellini, Maria Pietronilla Penna

**Affiliations:** 1Department of Pedagogy, Psychology, Philosophy, University of Cagliari, 09123 Cagliari, Italy; mirian.agus@unica.it (M.A.); penna@unica.it (M.P.P.); 2Institute of Education, Faculty of Philosophy, Jagiellonian University, 31-007 Kraków, Poland; lukasz.tomczyk@uj.edu.pl; 3Unit of Statistics, IRCCS, Centro San Giovanni di Dio Fatebenefratelli of Brescia, 25125 Brescia, Italy; nbonfiglio@fatebenefratelli.eu; 4Department of Political and Social Sciences, University of Cagliari, 09123 Cagliari, Italy; bell.diego@tiscali.it

**Keywords:** smartphone distraction, smartphone problematic use, online vigilance, attention impulsiveness, multitasking, emotion regulation, cognitive failures

## Abstract

This work aimed to validate the use of the Smartphone Distraction Scale (SDS) in Italy. The SDS was devised to assess distraction related to smartphone use in adult populations. A cross-sectional study was conducted among *n* = 609 adults (females = 76.4%; mean age = 30.26; SD age = 9.90). An assessment of the factorial structure of the Italian version was carried out using exploratory factor analysis (EFA) and confirmatory factor analysis (CFA). The four factors identified by Throuvala and colleagues were confirmed (i.e., attention impulsiveness, online vigilance, multitasking and emotion regulation). Reliability was assessed using Cronbach’s alpha coefficient (0.703–0.889). The scale’s scores showed significant linear correlations with validated instruments, including the Mobile Phone Problematic Use Scale (MPPUS)and the Cognitive Failures Questionnaire (CFQ). A multivariate analysis of variance showed significant differences in the means among participants belonging to different age groups (born before 1995 vs. born after 1996). In summary, the good psychometric properties observed led us to assume that this instrument can be applied and used in Italian studies to assess the cognitive dimension of distraction related to the use of smartphones.

## 1. Introduction

### 1.1. Smartphone Use and Overuse

From their entry into the market to the present, smartphones and their use have spread remarkably. Smartphone use is now so widespread that it is regarded as an extension of oneself and one’s body, having become an essential tool in individuals’ lives. Smartphones have numerous advantages, such as being helpful in managing activities of daily living. In addition to providing speed and fluidity in communication and exchanges between people, they can facilitate working life and productivity, as well as study and learning activities [1]. The advantages of smartphone use are certainly tied to the infinite content and material that can be accessed. Free access to the Web greatly increases the possibilities for entertainment and social media use, which increases the desire to use and spend as much time as possible with one’s smartphone [2].

The creative and appropriate use of a smartphone generates a state of deep concentration, which increases performance levels—an event called flow. When using such devices, the concept of temporality disappears, delimiting contact between only the individual and the smartphone.

One aspect that contributes to smartphone overuse is the sense of connection/social connection. This can be annulled or altered when one does not have access to the tool that enables it. In particular, the absence or presence of connections tied to social media use can result in the fear-of-missing-out (FoMO) phenomenon. FoMO is defined as a pervasive preoccupation with others having rewarding experiences from which one is absent and is characterised by the desire to always be connected (i.e., to know what other people are doing) [3,4,5,6,7]. Given that smartphones allow constant connections online, FoMO and smartphone overuse are closely related problematic phenomena.

Smartphone overuse and related factors are now being questioned urgently in the literature to understand which elements are related to such phenomena and which elements can be identified as precursors. With these data, preventive action can be taken before overuse turns into a true addiction.

Notably, while Bianchi and Phillips [8,9] found that extroversion and low self-esteem appear to be precursors to problematic smartphone use, many factors can contribute. For example, Busch and McCarthy’s review [10] of 239 studies identified macro areas that can be considered antecedents of this phenomenon, such as control (understood as the ability to control impulses and self-regulate the use of the smartphone), health (both physical and mental), factors linked to the family context and the technological characteristics of the object itself.

Various attempts to find models of the causes and effects of smartphone overuse and problematic smartphone use exist in the literature, one of the most comprehensive being the latest version of Billieux’s pathway model from 2015 [11]. According to the author, this behaviour can arise and be driven by three pathways. The pathway of over-reassurance is represented by an individual’s strong need to maintain relationships and obtain reassurance from others. Self-esteem is low, as is emotional stability, and the level of anxiety, especially social anxiety, is high. The impulsive–antisocial pathway is characterised by uncontrolled and unregulated impulses. Behaviour is uncontrolled and often aggressive. The extroversion pathway focuses on rewards derived from the outside, a strong and constant desire to communicate with others and establish new relationships, and strong sensitivity and dependence [12,13,14].

Among the aforementioned works, there is significant agreement regarding the variables that lead to the manifestation of problematic mobile phone use, and among these, the most frequently named is self-control, extending the concept to impulsivity, cognitive failures and dysfunctional emotional regulation. Cognitive failures refer to lapses or errors in cognitive functioning, such as memory lapses, attentional lapses and difficulties in concentration [15]. Of course, factors related to cognitive failures may vary based on many elements (e.g., frequency and intensity of smartphone use, individual susceptibility and the presence of pre-existing cognitive vulnerabilities). However, the literature shows a worrying finding—an increase in cognitive failures, even among younger people [16]. For instance, constantly relying on smartphones to retrieve information may help manage cognitive overload, but it may also reduce the brain’s ability to encode and store information and thus reduce memorisation capacity [17].

### 1.2. Smartphone Distraction

In the long term, smartphone overuse and compulsive use reduce concentration abilities, wellbeing and creativity [18,19]. The massive use of smartphones could affect cognitive functions and influence performance, especially in learning contexts, such as schools [20,21,22], and in the professional field [22,23].

Firth and colleagues [17] underlined how the constant use of the internet above all (thanks to the smartphone) can affect attentional abilities due to having one’s attention constantly divided. Memory processes can also be affected by the amount of information in storage and its retrieval, impacting social cognition. This is because social settings and environments are changed, and the concepts of social life, self-concepts and self-esteem are completely renewed. These findings on changes in cognitive processes and functions have been confirmed by several studies in the literature [24].

Liebherr and colleagues [25] found that smartphone overuse affects attention, inhibition and working memory if moderated by individual attributes and situational factors. In general, some studies have found a close link between smartphone overuse and distraction. Skowronek and colleagues [26] found that the mere presence of the smartphone reduced attentional performance and increased distraction. In defining distraction, it is difficult to find an unambiguous definition of this concept in the literature. The link between attention, concentration, cognitive overload and distraction is certainly evident. In general, the literature defines distraction as shifting from a primary task due to psychological reactions triggered by external stimuli or nonprimary-level activities [27].

Selective attention, defined as the ability to counteract distractions and focus attention on certain stimuli, is reduced over time so that less information is processed, suggesting a tendency towards economy or negative cognitive economy to achieve a goal [28]. Therefore, the smartphone is a multitasking tool that produces rewards and reinforcements at intervals, influencing and stimulating its gradual and/or constant use. Excessive use results in a permanent state of inattention [29].

Attention is a cognitive process that allows one to focus on certain stimuli to process them, but only a limited number of stimuli can be processed at any given time [30]. Therefore, if the demands of the environment require more attention than is available, the cognitive system becomes overloaded and performance suffers [31]. We live in the age of multitasking, which constantly tests our attentional capacity, and in an environment full of distractions, such as visual stimuli, constant notifications and a constant bombardment of information [29]. The causes of distraction are endogenous and exogenous in nature. Distraction has also been linked to functional emotion regulation strategies utilised to relieve emotional distress [32]. The level of distraction is often subjective and variable, but all the relevant literature agrees that the arrival of new technologies has changed individuals’ ways of relating to the environment, with a general increase in distraction levels. This is amplified by the fact that people’s lives now take place on two planes: the physical and the virtual.

### 1.3. Smartphone Distraction Scale

Throuvala and colleagues [32] presented a model of smartphone distraction and proposed an instrument called the Smartphone Distraction Scale (SDS) that could measure it. They highlighted four basic factors: attention impulsiveness, online vigilance, multitasking and emotion regulation, all of which include distracting attention from the current task, albeit with different motivations. For example, an item measuring the subscale of emotion regulation reads, “Using my mobile phone distracts me from negative and unpleasant thoughts”. In this case, the smartphone is used as a dysfunctional coping strategy, as the individual diverts attention from the negative situation without resolving it. The area of online vigilance encompasses distractions due to thoughts about activities that one might carry out online, especially those related to social networks, and it appears to be related to FoMO. Multitasking, as the term suggests, refers to performing several tasks at the same time, whether they are carried out through a smartphone (e.g., “I use several apps on my phone when I work”) or involve using a smartphone while engaged in other activities, such as walking. Finally, attention impulsiveness refers to the distraction produced by the smartphone itself as an object and thus the attention it attracts, which is linked to the subject’s low self-control [33].

SDS was validated in two other countries: in China, three factors were identified (i.e., attention impulsiveness, emotion regulation and multitasking) in a sample of undergraduates [34], and in Turkey [35], the four factors identified in the original work were confirmed (i.e., attention impulsivity, online vigilance, multitasking and emotion regulation).

In this work, we validated the Italian version of the SDS [32] related to smartphone use in the adult population.

## 2. Aims

Based on the literature and previous evaluations and due to the non-existence of a scale that assesses these aspects in the Italian context, this study aimed to validate the Italian version of the SDS [32]. The SDS is useful for measuring the application of cognitive strategies of distraction related to the use of technologies in adults.

Specifically, the psychometric features, validity and reliability of the SDS were evaluated by the application of exploratory factor analysis (EFA) and confirmatory factor analysis (CFA). Furthermore, the convergent and divergent validities were assessed by examining the relationship between the scale and other psychological dimensions (e.g., the problematic use of smartphones and cognitive failures), which were assessed by instruments not yet validated in Italy. Finally, criterion-related validity was considered in evaluating the differences in the average SDS regarding age groups.

## 3. Methods

### 3.1. Participants

Participants consisted of 609 Italian adults. Precisely, the sample included 465 females (76.4%), aged between 18 and 65 years old (mean age = 30.26; standard deviation age = 9.90).

On average, they spend 4.73 (±2.84) hours per day connected online on workdays and 4.33 (±2.67) hours per day on holidays. They largely use chats (mean = 3.68; SD = 1.03) and mainly use their smartphones to connect online (mean = 4.24; SD = 0.87). Table 1 showed in detail the descriptive statistics regarding the variables assessed in this study.

We applied a cross-sectional study with a non-probabilistic sample. The protocol of the research was administered in digital format using the platform SurveyMonkey to reach as many participants as possible.

It was distributed via email, via groups constituted in social networks.

The administration of the questionnaire was carried out from June 2022 to October 2022.

The local ethical committee of Cagliari State University approved the protocol of the research; the participants gave their informed consent for their participation and for the treatment of the data for scientific aims.

### 3.2. Procedure

The protocol of the study was organised into different sections.

The first one evaluated the classical sociodemographic aspects (age, gender, education) and the habits related to the use of technological instruments (daily hours spent online on workdays and holidays, instruments used to connect online).

### 3.3. Distraction

The Smartphone Distraction Scale (SDS) is a self-assessment scale recently devised by Throuvala and colleagues [32] to evaluate distraction in relation to smartphone use as a functional emotion regulation approach employed to mitigate emotional distress.

It evaluates 16 items on a five-point Likert scale (varying from 1—almost never—to 5—almost always), identifying 4 dimensions, each having 4 items: attention impulsiveness (item example: “I get distracted by my phone apps”), online vigilance (item example: “I think a lot about checking my phone when I can’t access it”), emotion regulation (item example: “Using my phone distracts me from negative or unpleasant thoughts”) and multitasking (item example: “I can easily follow conversations while using my phone”).

In order to assess the psychometric proprieties of the Italian version of the SDS, we applied the classical procedures identified in the literature [36,37]. Initially, three independent scholars translated the SDS from English to Italian; subsequently, they reached an agreement on a joint version. This version of the SDS was then back-translated by a bilingual researcher who has broad expertise in psychological studies. As a final step, the SDS items of the Italian version were evaluated by three experts in the problematic use of smartphones.

### 3.4. CFQ

The Cognitive Failures Questionnaire [15] was administered in its validated Italian version [38]. This self-assessment instrument (comprising 25 items, evaluated on a 5-point Likert scale, ranging from 0 to 4) explores the occurrence of trivial errors and cognitive failures that are expected to occur to everyone in everyday life regarding different facets (perceptual area, memory and motor control). The reliability of the Italian version regarding the general factor defined as “CFQ—Perceived cognitive failure” was 0.81. Some items, for example, are the following: ‘‘Do you find you forget appointments?” “Do you start doing one thing at home and get distracted into doing something else (unintentionally)?” ‘‘Do you read something and find you haven’t been thinking about it and must read it again?’’

### 3.5. MPPUS

The problematic use of smartphones was evaluated by the administration of the Mobile Phone Problematic Use Scale (MPPUS) [8] in its Italian version [9]. This instrument (characterised by 24 items, assessed on a Likert scale ranging from 1 to 5) evaluated the following dimensions: Withdrawal and social aspects, Craving and escape from other problems and the General dimension of problematic use of smartphone.

### 3.6. Data Analysis

Data processing was carried out with JASP software, version 0.16.3.0 (https://jasp-stats.org/accessed on 15 May 2023) [39]. To analyse the data, the participants were randomly divided into two sub-samples. The first was used to apply EFA (Stage 1, *n* = 305), and the second was used to apply CFA (Stage 2, *n* = 304). For the assessment of psychometric features and construct validity, EFA was applied in Stage 1. CFA was carried out in the second random subsample (Stage 2). In CFA, we applied diagonally weighted least squares (DWLS) estimation with the robust method of estimation, which is helpful in the assessment of ordinated categorical variables (e.g., Likert scales) [40]. To assess the goodness of fit in the CFA, we referred to the classical indices and rules of thumb defined in the literature [41,42,43]. Because chi-squared (Chi2) was affected by the number of participants, the ratio of Chi2 to their degrees of freedom (Chi2/df) was evaluated (considered adequate when smaller than three) [43]. Concerning the root mean square error of approximation (RMSEA), this statistic is considered suitable when it is between 0.05 and 0.08 [41,42,43]. We also computed the confidence interval at 90%. The comparative fit index (CFI) and the Tucker–Lewis index (TLI) were calculated; their values should be greater than 0.90 to be considered acceptable [41,42]. Furthermore, the standardised root mean square residual (SRMR) was computed, considering a cut-off of <0.08 [42]. The analysis of internal consistency was carried out using Cronbach’s alpha coefficient; the criterion and concurrent validities were assessed by Pearson’s linear correlation between the SDS and other measurements (number of hours spent online and psychological dimensions measured by the MPPUS and CFQ instruments). Furthermore, criterion-related validity was considered based on the potential differences between participants belonging to different generations (Generation Z versus older individuals) through a multivariate analysis of variance (MANOVA) [44,45].

## 4. Results

### 4.1. Stage 1—EFA

EFA was carried out using the maximum likelihood extraction method and the Promax rotation of factors. Regarding the assumption’s checks, the findings suggest that the sample was adequate and that EFA could be applied (Bartlett’s test of sphericity = 2367.0, df = 120; *p* < 0.001; overall KMO = 0.897). The graphical scree plot examination and parallel analysis (testing whether a factor identification might be due to chance) were applied to identify an adequate number of factors to retain in this questionnaire [46,47] (see Appendix A). The analyses indicated that the solution was suitable with the four factors suggested by the authors of the scale [32] (attention impulsiveness, emotion regulation, online vigilance and multitasking). The EFA results corresponded to those originally obtained by the authors of the instrument (Table 2). Specifically, the factor attention impulsiveness was saturated by items 1, 2, 3 and 4; online vigilance showed high levels of saturation for items 5, 6, 7 and 8; multitasking highlighted high factor loadings regarding items 9, 10, 11 and 12; and emotion regulation was saturated by items 13, 14, 15 and 16 (Table 2). All four factors showed good indices of Cronbach’s alpha reliability (ranging from 0.703 to 0.889), highlighting high levels of internal consistency for each factor. The details of the statistics related to EFA are shown in Table 2.

### 4.2. Stage 2—CFA

CFA was applied with the DWLS estimation. The assessment focused on three competing models that were identified based on the work of the authors of the original instrument and on the literature. Model 1 assessed the fit of the one-factor solution, and Model 2 evaluated the first-order solution, which was characterised by four correlated factors. These were devised by the authors of the SDS and identified by the previous EFA in Subsample 1; Model 3 estimated the fit of a second-order factor over the four factors identified by the authors and in the previous EFA (see Appendix A) The appropriate CFA model was chosen based on the evaluation of a variety of features, as suggested by the literature [48]. Specifically, the following aspects were considered: the number of free parameters to be estimated, the correlations between the factors, the fit indices, the parsimony of the model and the model propensity [49]. The latter feature, which is defined as model flexibility [50] or model complexity, concerns “the ability of a model to fit a diverse array of data patterns well” [49]. This attribute (called overfitting) can compromise the ability of a CFA model to be generalised to new samples extracted from the same population [49]. Parsimonious models restrict model-compatible data and are not as susceptible to overfitting [51]. Hence, to ensure high generalisability, it is desirable to find a balance between the goodness-of-fit assessment and propensity by selecting the model that provides the best description of the data in the least complex way [48]. Regarding these aspects, in the CFA application, the goodness-of-fit indices of RMSEA and TLI, accounting for the number of free parameters, allow for the application of fair-minded model selection choices [48]. Table 3 illustrates the indices of the goodness of fit of our competing models. Model 1 shows low indices of goodness of fit to the data (regarding the rules of thumb in the literature [41]). Models 2 and 3 show, respectively, good and similar indices of goodness of fit; thus, after applying the above-mentioned evaluations, we chose Model 2 for CFA, characterised by a first-order solution with four correlated factors (see Table 3 and Figure 1). Model 2 confirmed the factorial structure identified in the previously applied EFA and by the original authors of the instrument, with four items for each factor.

In the total sample, to assess the criterion and concurrent validity, we evaluated Pearson’s linear correlation between the SDS scores and the other variables assessed in the protocol, specifically, the number of hours spent online daily (workdays and holidays) and the scores regarding the validated instruments administered in the protocol (Table 4). We observed significant positive linear correlations between all variables, except for the SDS dimension of online vigilance, which did not show a significant linear correlation with the number of hours spent online daily (holidays and workdays). These data suggest the presence of direct relationships, highlighting that there are strict positive associations between all four factors of the SDS, but also between the SDS factors and the dimensions inquiring about the problematic use of smartphones (assessed by the MPPUS scales, such as withdrawal, social aspects, cravings and escaping from other problems) and between the SDS factors and aspects related to cognitive failures (evaluated by the CFQ) (Table 4). These data corroborate the validity of the SDS in the Italian version.

Furthermore, to assess the criterion-related validity, a MANOVA was applied to the total sample using the generation of participants (born before 1995 or born after 1996) as a factor and the scores for the four factors of SDS (attention impulsiveness, emotion regulation, online vigilance and multitasking) as dependent variables.

The multivariate Wilks’ test highlighted a significant effect (Wilks’ Lambda = 0.896; df = 4; 581; *p* < 0.001; partial eta squared = 0.104). The univariate tests confirmed significant differences in the means of the two generations for the four factors of SDS. For all factors, the participants born after 1996 obtained significantly higher mean scores than the older participants (see Table 5 and Table 6).

## 5. Discussion

Smartphone use can lead to a series of advantages, but excessive use can bring with it many disadvantages and problems, such as negative effects on cognitive processes. Many studies have confirmed that smartphone overuse is associated with depression and anxiety disorders [52], psychological and physical health problems [53,54], ocular diseases [55], musculoskeletal problems [56] and general dysfunctions in daily life activities [56,57]. Other studies have affirmed the presence of specific problems, such as FoMO on social media [57], nomophobia (i.e., fear of being away from your smartphone) [58], ringxiety (i.e., hearing non-existent smartphone notifications) [59], vamping [24,60] (i.e., using a smartphone for long periods of the night), phubbing (i.e., social dissociation by constantly consulting with a smartphone) [61] and smartphone addiction (i.e., a disorder with serious effects on physical and psychological health) [53,62]. Smartphone overuse influences the perception of time, academic and work achievements and social interactions. People using smartphones exhibit decreased concentration and attention on other activities, tasks or interactions that occur in the real world. Smartphone overuse is strictly tied to distraction, a phenomenon that can negatively impact individuals’ lives [16].

Distraction is not easy to define; the literature offers various facets of the concept. There is also ambiguous agreement on the causes and effects of distraction that probably depend on complex interactions between factors, such as the attentional demands of the distracting information, the nature of the primary task and the similarity of operations between those required by the primary task and those needed to handle the distracting information [63]. The literature shows that the increase seen in distraction levels is mainly due to smartphone use and overuse, with smartphones becoming super-stimuli. It is not possible to inhibit the response because immediacy takes over in the response and cannot be deferred to another time. It is also not possible to focus on the task because there is an overload of short-term memory and the attention system.

The stimulus takes up space, and whatever has the strongest emphasis receives attention. This generates constant distraction from the main task. In general, distraction can affect cognitive performance and, importantly, memory and attention [63]. For example, many studies have demonstrated the effects of distraction on driving or in work contexts that require a maximum level of attention [64,65,66]. Being distracted involves being absorbed in virtual activities that divert attention from the present moment and the surrounding environment. As seen among the dimensions of the SDS tool, smartphone distraction can be related to a wide variety of factors. There can be frequent interruptions from incoming messages, notifications or alerts from various apps; the constant need to check and respond immediately; the desire for constant control over what is happening online; attempts to multitask, such as texting while walking, checking email during conversations or using social media while studying or working; and the use of a smartphone as a tool for emotional regulation or dysregulation

What is also clear from our analysis is the close link between distraction and possible cognitive failure, which is due to many elements, including information overload and a continuous influx of data [48]. Our data confirm what the recent literature on the subject states, namely, that cognitive decline is especially high among the youngest. Descriptive statistics confirm a mean difference in the SDS dimensions regarding age; however, the participants born in 1996 or later obtained significantly higher mean scores than the older participants involved in the study. The same results were found in previous studies [16]. A key element to reflect on from this and previous research is the need to understand the cognitive and social effects of internet and smartphone overuse. It is crucial that future studies understand their effects on a person’s cognitive development. Therefore, studies should include longitudinal monitoring and make comparisons between young and older participants.

## 6. Conclusions

Careful management of smartphone use and the adoption of strategies to ensure its functional use can help minimise distractions and avoid negative impacts on life activities and cognitive functions. In this regard, it would be interesting to longitudinally study the possible effects of excessive smartphone use on the reduction in cognitive reserve, which is especially crucial for successful ageing [67]. Some studies have even referred to digital dementia among young people, referring mainly to the effects of excessive screen time [68,69].

Generally, we can state that smartphone distraction can be a danger for young people, not only because of the possibility of cognitive impairment but also regarding the negative impact on school performance and self-regulation during study time, an increased risk of accidents, reduced physical activity and social and emotional implications.

It is important to emphasise that studies such as ours do not intend to demonise smartphone use but aim to stress that excessive and uncontrolled use carries risks. In light of what has been read and observed, one can consider the need to educate individuals as early as adolescence on the conscious use of smartphones, which has great potential but can also have negative aspects. Studies should therefore include longitudinal monitoring and also make comparisons between young and old individuals. Furthermore, such studies should be extended to various social and life contexts.

It is crucial to have tools such as the SDS at our disposal to allow the easy and immediate identification of risk factors. The scale in this form could be used in educational and research contexts to measure specific aspects related to distraction in an efficient and meaningful manner.

## Figures and Tables

**Figure 1 ijerph-20-06509-f001:**
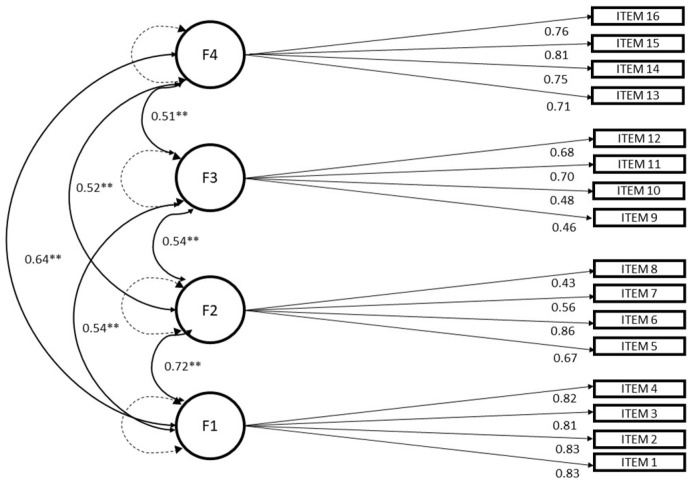
CFA, path diagram of Model 2. Note: F1—attention impulsiveness F2—online vigilance; F3—multitasking; F4—emotion regulation (all paths are significant, *p* < 0.01 **).

**Table 1 ijerph-20-06509-t001:** Descriptive statistics of the variables assessed.

	Fr (%)	Mean (SD)
Age		30.26 (9.90)
Gender—Female fr (%)	465 (76.4)	
Education:		
High school	540 (88.67)	
Degree or postgraduate certification	69 (11.33)	
		Mean (SD)—Range from 0 to 18
How many hours on average do you spend per day online on working days ?		4.73 (2.84)
How many hours on average do you spend per day online on holidays ?		4.33 (2.67)
How often do you use the following digital tools?		Mean (SD)—Range from 1 to 5
(Social networks)		3.41 (1.01)
(Platforms for teaching/work activities)		2.53 (1.04)
(Chat)		3.68 (1.03)
(Blog)		1.48 (0.70)
(Forum)		1.46 (0.67)
(Virtual Worlds)		1.30 (0.67)
(Online games)		1.53 (0.89)
(Mail)		3.02 (0.97)
(Photo shopping)		1.53 (0.82)
(Other)		2.81 (1.44)
How often do you use each of the following devices to connect to the internet?		Mean (SD)—Range from 1 to 5
(Cell phone)		4.24 (0.87)
(Tablet)		1.87 (1.12)
(PC)		3.18 (1.15)
(Smart tv)		2.02 (1.05)
(Other)		1.11 (0.41)
MPPUS		Mean (SD)
General factor		49.32 (15.61)
Withdrawal and social aspects		13.33 (4.84)
Craving and escape from other problems		28.55 (9.43)
CFQ		
Perceived cognitive failure		38.71 (16.52)
SDS		
Attention impulsiveness		10.17 (4.19)
Emotion regulation		9.84 (4.14)
Online vigilance		6.41 (2.70)
Multitasking		9.21 (3.24)

**Table 2 ijerph-20-06509-t002:** Factor loadings for EFA: “Maximum likelihood” extraction method with Promax rotation.

	Factor Loadings			
Items (Italian Version)	1 Attention Impulsiveness	2 Emotion Regulation	3 Online Vigilance	4 Multitasking
Item1: I get distracted by my phone notifications [Mi distraggo a causa delle notifiche del mio telefono]	0.836	−0.019	−0.070	0.049
Item 2: I get distracted by my phone apps [Mi distraggo a causa delle applicazioni del mio telefono]	0.847	−0.053	−0.099	0.073
Item 3: I get distracted by just having my phone next to me [Mi distraggo anche solo avendo il telefono vicino]	0.837	−0.078	0.061	−0.042
Item 4: I get distracted by my phone even when my full attention is required on other tasks [Vengo distratto/a dal mio telefono anche quando la mia attenzione dovrebbe essere completamente diretta verso altri compiti]	0.877	0.039	0.021	−0.101
Item 5: I get anxious if I don’t check messages immediately on my phone [Divento ansioso/a se non controllo immediatamente i messaggi sul mio telefono]	0.160	0.191	0.653	−0.193
Item 6: I think a lot about checking my phone when I can’t access it [Penso molto a controllare il telefono quando non posso usarlo]	0.236	0.067	0.578	−0.020
Item 7: I get distracted with what I could post while doing other tasks [Mi distraggo mentre svolgo altri compiti pensando a cosa potrei postare]	−0.089	−0.076	0.833	0.119
Item 8: I get distracted thinking how many likes and comments I will get while doing other tasks [Mi distraggo mentre svolgo altri compiti pensando a quanti like e commenti riceverò]	−0.117	−0.061	0.755	0.052
Item 9: I use several applications on my phone while working [Mentre lavoro, uso diverse applicazioni sul mio telefono]	0.239	−0.003	0.161	0.319
Item 10: I can easily follow conversations while using my phone [Riesco a seguire facilmente una conversazione mentre sto usando il telefono]	−0.060	0.061	−0.032	0.543
Item 11: I often walk and use my phone at the same time [Spesso cammino e uso il telefono contemporaneamente]	0.085	0.066	−0.059	0.635
Item 12: I often talk to others while checking what’s on my phone [Spesso parlo con gli altri e contemporaneamente controllo il mio telefono]	−0.023	−0.089	0.142	0.686
Item 13: Using my phone distracts me from doing unpleasant things [Usare il telefono mi distrae dal fare cose spiacevoli]	−0.173	0.868	0.011	0.052
Item 14: Using my phone distracts me from negative or unpleasant thoughts [Usare il telefono mi distrae da pensieri negativi e sgradevoli]	−0.120	0.938	0.029	−0.026
Item 15: Using my phone distracts me from tasks that are tedious or difficult [Usare il telefono mi distrae da compiti noiosi o difficili]	0.140	0.645	−0.057	0.134
Item 16: Using my phone distracts me when I’m under pressure [Usare il telefono mi distrae quando sono sotto pressione]	0.178	0.670	−0.029	−0.045
Eigenvalue	5.989	1.127	0.7578	0.487
Percentage of variance explained	19.520	15.940	13.490	8.940
Cronbach’s alpha	0.889	0.869	0.825	0.703

**Table 3 ijerph-20-06509-t003:** Indices of goodness of fit for Model 1, Model 2 and Model 3.

	Model 1 Single-Factor CFA	Model 2 First-Order CFA Four Factors	Model 3 Second-Order CFA One General Factor and Four Factors
Chi-squared	637.783 df = 104 *p* < 0.001	231.886 df = 98 *p* < 0.001	235.979 df = 100 *p* < 0.001
Chi-squared/df	6.132	2.3659	2.3597
RMSEA	0.133	0.068	0.068
RMSEA 90% CI	0.123–0.143	0.057–0.080	0.057–0.080
CFI	0.731	0.932	0.931
TLI	0.689	0.917	0.918
SRMR	0.093	0.058	0.059

Note: df = degree of freedom; RMSEA (90% CI) = root mean square error of approximation with confidence interval; CFI = comparative fit index; TLI = Tucker–Lewis index; SRMR = standardised root mean square residual.

**Table 4 ijerph-20-06509-t004:** Pearson’s r linear correlations between assessed variables.

Variable		1		2		3		4		5		6		7		8		9
1. MPPUS General factor	r	—																
	*p*	—																
2. MPPUS Withdrawal and social aspects	r	0.805	**	—														
	*p*	<0.001		—														
3. MPPUS Craving and escape from other problems	r	0.943	**	0.601	**	—												
	*p*	<0.001		<0.001		—												
4. General factor CFQ	r	0.518	**	0.439	**	0.448	**	—										
	*p*	<0.001		<0.001		<0.001		—										
5. SDS_attention impulsiveness	r	0.679	**	0.451	**	0.620	**	0.416	**	—								
	*p*	<0.001		<0.001		<0.001		<0.001		—								
6. SDS_online vigilance	r	0.660	**	0.470	**	0.664	**	0.319	**	0.591	**	—						
	*p*	<0.001		<0.001		<0.001		<0.001		<0.001		—						
7. SDS_multitasking	r	0.426	**	0.228	**	0.440	**	0.221	**	0.425	**	0.369	**	—				
	*p*	<0.001		<0.001		<0.001		<0.001		<0.001		<0.001		—				
8. SDS_ emotion regulation	r	0.591	**	0.371	**	0.597	**	0.391	**	0.543	**	0.423	**	0.396	**	—		
	*p*	<0.001		<0.001		<0.001		<0.001		<0.001		<0.001		<0.001		—		
9. How many hours on average do you spend per day online on working days?	r	0.215	**	0.180	*	0.174	*	0.179	*	0.176	*	0.062		0.252	**	0.112		—
	*p*	<0.001		0.002		0.003		0.003		0.003		0.293		<0.001		0.056		—
10. How many hours on average do you spend per day online on holidays?	r	0.299	**	0.266	**	0.259	**	0.238	**	0.229	**	0.062		0.208	**	0.253	**	0.654
	*p*	<0.001		<0.001		<0.001		<0.001		<0.001		0.294		<0.001		<0.001		<0.001

* *p* < 0.01, ** *p* < 0.001.

**Table 5 ijerph-20-06509-t005:** Univariate effects for the MANOVA.

Factor	Dependent Variable	Df(b; w)	F	*p*	Partial Eta Squared	Power Observed
Generation	SDS_ attention impulsiveness	1; 584	53,978	<0.001	0.085	1.000
	SDS_online vigilance	1; 584	6754	0.010	0.011	0.737
	SDS_ multitasking	1; 584	26,427	<0.001	0.043	0.999
	SDS_ emotion regulation	1; 584	22,525	<0.001	0.037	0.997

**Table 6 ijerph-20-06509-t006:** Descriptive statistics regarding mean differences in the SDS dimensions in relation to the level of age (born before 1995 vs. born after 1996).

	Generation	Mean	SD	N
SDS_F1_attention impulsiveness	Born before 1995	8.983	3.823	301
	Born after 1996	11.425	4.218	285
	Total	10.171	4.198	586
SDS_F2_online vigilance	Born before 1995	6.130	2.551	301
	Born after 1996	6.709	2.843	285
	Total	6.411	2.710	586
SDS_F3_multitasking	Born before 1995	8.562	3.119	301
	Born after 1996	9.912	3.242	285
	Total	9.218	3.248	586
SDS_F4_emotion regulation	Born before 1995	9.070	4.116	301
	Born after 1996	10.667	4.023	285
	Total	9.846	4.145	586

## Data Availability

The datasets for this study are available from the corresponding author on reasonable request.

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
