# Peer review of "Smartphone Distraction: Italian Validation of the Smartphone Distraction Scale (SDS)"

_ijerph, 2023, doi:10.3390/ijerph20156509_

Round 1

Reviewer 1 Report

Overall English needs to be revised example line 91 (misspelled than).

Other than that is an interesting scale that will help the population in reviewing the effects that a smartphone can cause in any context. 

This study can also be tailored to specific groups of professionals that this smartphone distractions can cause serious consequences.

English should be revised since there are various grammatical errors overall the manuscript as I mentioned before for example line 91, "than" is misspelled. 

Author Response

Dear Editor and Referees,

Thank you for your letter, Manuscript ID: ijerph-2428233-R1, entitled “Smartphone distraction: Italian Validation of the Smartphone Distraction Scale (SDS)”, and for giving us the opportunity to review and resubmit the paper.

We are very grateful to your and the reviewers’ comments and suggestions; we are deeply appreciative of your careful reading.

Detailed replies to your comments are enumerated below, with the list of modifications and integrations. We hope this revised version now satisfies the requirements for publication in your journal.

Then, we submit the revised version of paper; for clarity new portions, added or modified in response to the referees’ comments, are highlighted in the manuscript; furthermore, the tracked version of the manuscript is attached.

Thank you very much.

The Authors

Reviewer 2 Report

Overall, this paper is a well-written and informative contribution to the study of psychometric properties of the Italian version of the Smartphone Distraction Scale (SDS). The authors have provided a clear and concise overview of the SDS. The aims of the study are well-formulated, and the study design and data analysis appear to be rigorous and appropriate. This paper makes an important contribution to the literature that SDS is a useful measurement tool in other cultures.   

My main comments and criticism of the paper is as follows:

I think that in the introduction it would be useful to write a few sentences about the results of the factor analysis of SDS in different countries (e.g. Zhao et al., 2022; Bilge et al., 2022).

The number of factors is somewhat overstated, it is not clear how many factors are likely to have been identified by, for example, the parallel analysis. Based on the eigenvalues (Table 3), it is not more than 2. A justification should be given as to why a CFA with 4 factors was performed.

The comparison of first order and second order models is adequate, but it would be worthwhile to show other competing model fits: e.g. 1-factor model, 2 or 3-factor models. In addition, the second order model or the bifactor model may be of interest because it can be used to support the use of the main scale score. Since the correlation between the factors is not very high, the choice between first order and second order models is an important choice.

In Figure 2, the mean factor score is shown as a floating point on the axis; these are rather scale scores.

I was pleased to read your well structured and precisely written manuscript. In my opinion, the additional notes and analysis mentioned above would greatly enhance the value of the paper.

References

Bilge, Y., Bilge, Y., & Sezgin, E. (2022). Turkish Adaptation of the Smartphone Distraction Scale (SDS). Current Approaches in Psychiatry/Psikiyatride Guncel Yaklasimlar, 14.

Zhao, X., Hu, T., Qiao, G., Li, C., Wu, M., Yang, F., & Zhou, J. (2022). Psychometric properties of the smartphone distraction scale in Chinese college students: validity, reliability and influencing factors. Frontiers in psychiatry13.

Author Response

(The authors gave the same response as above.)

Reviewer 3 Report

This study is an important contribution to the studies in smartphone distraction. It validated an Italian version of the SDS. I have several comments.

1.      I suggest the authors to check the latest version of Billieux's pathway model in 2015. There were three pathways. The link between their pathway model and smartphone distraction should be further addressed.

2.      In section 1.2, the studies about smartphone distraction and smartphone overuse or problematic smartphone use or smartphone addiction is not enough, The authors need to review more recent empirical studies in this topic. The authors seem to reply too much on Throuvala et al.’s study.

3.      The definition of smartphone distraction should be addressed clearly with references.

4.      Sections 3.3 should be measurements including all the scales. Section 4. Data Analysis should be included in the Methods chapter.

5.      Could you please revise the “4. Data Analysis” and avoid short paragraphs with only one or two sentences.

6.      Table 3 includes very importation information but was poorly presented.

7.      Figure 1 and 2 are very difficult to read. Please make sure that all the numbers and variable names are clearly presented. Meanwhile, the significances of the paths are missing.

8.      Table 5 was also not presented well.

9.      The discussion looks too short and comparisons with previous studies needs improvement. For example, the similarities and differences between this study and other studies on the validation of the SDS in different cultures. Please check if the SDS has been validated in other languages besides Italian.

Author Response

(The authors gave the same response as above.)

Round 2

Reviewer 3 Report

 Accept in present form